# EFFECTS OF SINGLE-ATTRIBUTE CONTROL ON THE MUSIC GENERATED BY FIGARO

**Rafik Hachana**
Innopolis University, Innopolis, 420500, Russia
`r.hachana@innopolis.university`

**Adil Mehmood Khan**
Innopolis University, Innopolis, 420500, Russia
`a.khan@innopolis.ru`
School of Computer Science, University of Hull,
HU67RX, Hull, UK
`a.m.khan@hull.ac.uk`

## ABSTRACT

We have experimented with controlling the musical attributes of the music generated by the FIGARO model, while evaluating the success of control and the overall musical quality. The results suggest non-trivial correlations between the musical attributes and the musical quality metrics.

## 1 INTRODUCTION

In this study, we attempt controlling the attributes of the generated music using the state-of-the-art model FIGARO, proposed by von Rütte et al. (2022). We aim to answer the following questions:

- How do different musical attributes respond to control?
- Which musical features affect the musical quality more when controlled?

## 2 RELATED WORK

The current SOTA models for music generation use the Music Transformer (Huang et al., 2018), and generate music in MIDI (Loy, 1985). Conditional generation can be done using transfer learning (Young et al., 2022), or sampling latent spaces (Shih et al. (2022), Ens & Pasquier (2020)) which allows global control over the generated sequence. Wu & Yang (2022) and Hadjeres & Crestel (2020) propose methods for fine-grained control i.e. the controlled attributes may vary over time.

The subject of this study is the FIGARO model, proposed by von Rütte et al. (2022). It allows the fine-grained control of music by using a description sequence representation that gets converted to the output sequence using a Transformer. The *description* is the combination of a learned latent representation of the input REMI-tokenized sequence, and a "domain expert" description with hand-crafted features. The FIGARO study did not experiment with altering individual attributes in the description sequence, hence the motivation behind our work. Further details about FIGARO are in Appendix B.

## 3 METHODOLOGY

We generate music sequences with the `figaro_expert` checkpoint that uses only the expert description function (no latent representation). The generated sequences come from descriptions sampled from the Clean MIDI Subset of the LMD dataset, with an altered attribute in the controlled cases. We used MIDI files of real songs in order to be consistent with the distribution of a real music dataset. We then evaluate the extent to which the attribute control was achieved, and how did the attribute control affect the musical quality.

For each input description, we create 6 extra altered descriptions by altering one attribute at a time. For the Instruments attribute, we randomly remove one instrument from the whole sequence. For the Chords, we transpose all the chords of the sequence. For the Mean Pitch, Mean Duration,

Table 1: Correlation between the attribute control amount and the evaluation metrics for Chords, Pitch, Velocity, Duration, Note Density, Pitch Histogram Entropy, Groove Similarity and Chord Progression Irregularity (Metrics explained in Appendix D)

| Control | Ch | P | V | D | ND | H1 | H4 | GS | CPI |
|---|---|---|---|---|---|---|---|---|---|
| Chords | 0.01 | -0.31 | 0.03 | -0.09 | 0.24 | -0.12 | -0.12 | 0.01 | -0.16 |
| Mean pitch | -0.30 | 0.23 | 0.02 | -0.01 | 0.08 | 0.08 | 0.02 | 0.46 | -0.17 |
| Mean velocity | 0.01 | 0.04 | -0.12 | 0.10 | -0.25 | 0.74 | 0.75 | 0.01 | -0.13 |
| Mean duration | 0.20 | 0.02 | -0.24 | -0.07 | -0.03 | 0.65 | 0.65 | 0.21 | 0.30 |
| Note density | -0.14 | -0.01 | 0.12 | 0.00 | -0.02 | -0.83 | -0.87 | -0.87 | 0.91 |

Mean Duration and Note Density, we shift their value by a uniformly-random delta over the whole sequence. (Appendix C)

We evaluate the generated music with the metrics of the FIGARO study for attribute fidelity, and with 3 metrics of MusDr (Wu & Yang, 2020) for quality (Appendix D). All code, data and generated samples are in our GitHub repository [1].

## 4 RESULTS AND DISCUSSION

The mean values of the evaluation metrics are in table 2. We notice the expected drop in attribute fidelity for the controlled attribute, however the quality metrics don't seem to drastically vary, which suggests that the quality of FIGARO's output is consistent when controlled. Table 1 shows that some attributes are correlated with the metrics of other attributes e.g. controlling the pitch is negatively correlated with the chords f1-score. The control of certain attributes is also correlated with quality metrics, which might not seem related to the attribute e.g. controlling Velocity or Duration correlates with a change in the pitch histogram entropy. The correlations suggest a learned association that the FIGARO model has learned. Finally, figure 1 shows the response of each attribute's metric to control. Mean pitch and Velocity are harder to control with bigger shifts, while Duration is more controllable if decreased. Chord transposition does not seem to affect the f1-score of chords which is low to begin with.

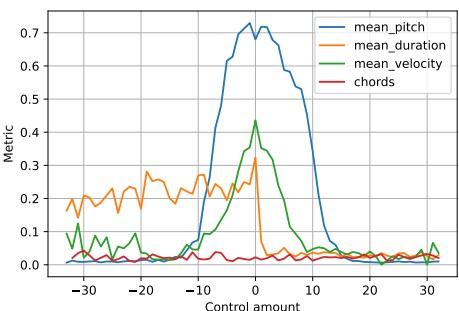

Figure 1: Response of the attribute-specific evaluation metrics to attribute control (More is better)

## 5 CONCLUSION

We can conclude that FIGARO maintains a consistent music quality when individual musical attributes are controlled, and that it has learned non-trivial correlations between these attributes. Certain attributes are less responsive to control (such as chords), which can be further understood with probe or ablation studies.

---

[1] https://github.com/RafikHachana/controlled-figaro

ACKNOWLEDGEMENTS

This research has been financially supported by the Analytical Center for the Government of the Russian Federation (Agreement No. 70-2021-00143 dd. 01.11.2021, IGK 000000D730321P5Q0002)

URM STATEMENT

The authors acknowledge that at least one key author of this work meets the URM criteria of ICLR 2023 Tiny Papers Track.

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

## A  FURTHER NUMERICAL RESULTS

Table 2: The mean value for each evaluation metric for each controlled attribute. The metrics are the Note Density NRMSE (ND), the Pitch OA as described in the FIGARO study (P), Duration OA (D), Velocity OA (V), Instrument f1-score (Instr), Chords f1-score (Ch), the pitch histogram entropy for 1 bar as implemented by the MusDr tool (H1), the pitch histogram entropy for 4 bars (H4), the grooving similarity across the track (GS), and the Chord Progression Irregularity (CPI). Details in Appendix D

| Attribute control | ND | P | D | V | Instr | Ch | H1 | H4 | GS | CPI |
|---|---|---|---|---|---|---|---|---|---|---|
| None | 0.50 | 0.73 | 0.40 | 0.45 | 0.95 | 0.56 | 1.87 | 2.32 | 0.98 | 0.71 |
| Instrument | 0.63 | 0.69 | 0.37 | 0.41 | 0.87 | 0.44 | 1.72 | 2.16 | 0.98 | 0.71 |
| Chords | 0.27 | 0.72 | 0.40 | 0.44 | 0.95 | 0.03 | 1.86 | 2.33 | 0.98 | 0.71 |
| Pitch | 0.65 | 0.18 | 0.33 | 0.39 | 0.93 | 0.50 | 1.74 | 2.17 | 0.98 | 0.69 |
| Velocity | 1.11 | 0.69 | 0.10 | 0.08 | 0.87 | 0.50 | 1.79 | 2.24 | 0.98 | 0.71 |
| Duration | 0.34 | 0.72 | 0.12 | 0.44 | 0.94 | 0.52 | 1.81 | 2.25 | 0.98 | 0.70 |
| Note density | 17.39 | 0.64 | 0.21 | 0.37 | 0.87 | 0.36 | 1.35 | 1.68 | 0.98 | 0.79 |

## B  THE FIGARO MODEL

This appendix describes the FIGARO model (von Rütte et al., 2022), its architecture and how it generates music.

### B.1  MODEL ARCHITECTURE

As depicted in figure 2, the FIGARO model is trained to translate a high-level music description sequence into a sequence of tokens that represents symbolic music using a seq2seq Transformer model. The high-level description sequence is calculated by combining a latent representation that is learnt jointly using a VQ-VAE, with the embedded tokens of the "Expert Description". The Expert Description is a sequence of tokens that represent handcrafted features of the music. This description is calculated from REMI symbolic music data (Ren et al., 2020) using the algorithm illustrated in Figure 4. The original FIGARO model was trained to reconstruct an input music sequence using only its high-level description. For the purpose of our study, we are using a FIGARO checkpoint that only uses the Expert description and no VQ-VAE. This enables us to generate Expert Descriptions from scratch and alter them freely.

### B.2  EXPERT DESCRIPTION

The FIGARO description is a sequence that describes the high-level features of a music piece. The description can be interpreted as the concatenation of multiple subsequences, which each subsequence describing one music bar and starting with a numbered bar token. A music bar is a unit of time in music theory. An example of a description sequence is shown in Figure 3.

The vocabulary of this sequence is comprised of the following token types:

- Bar tokens: The bar tokens denote the start of a new musical bar or measure.

- Time signature tokens: Denote the time signature of the current bar i.e. the duration of the bar in terms of musical beats.

- Instrument tokens: These tokens list all of the music instruments used in the current bar e.g. having `Instrument_Drums Instrument_Piano` means that the music in this bar is played by two instruments: Drums and Piano.

- Chord tokens: These tokens describe the chords played during the current bar. Chords are groupings of musical notes that give a high-level description of the music being played.

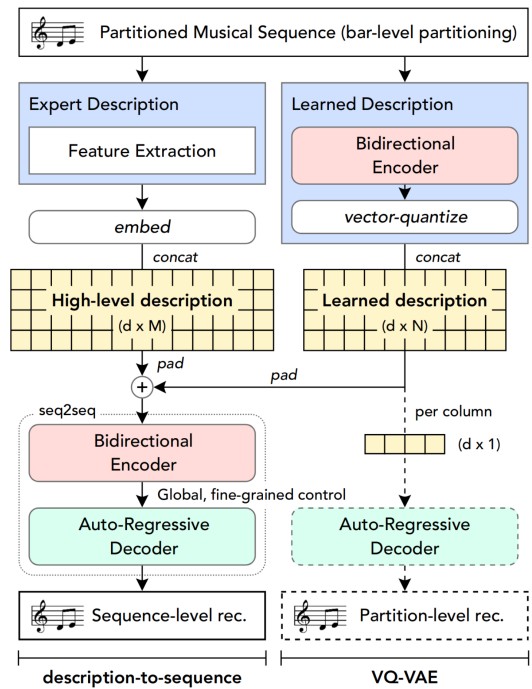

Figure 2: The architecture of the FIGARO model, taken from von Rütte et al. (2022)

```
<bos>
Bar_1  TimeSignature_4/4  NoteDensity_3
   MeanPitch_14  MeanVelocity_19  MeanDuration_32
   Instrument_Drums  Instrument_Piano
   Instrument_E-Piano  Instrument_SlapBass
   Chord_E:maj  Chord_F#:min7
Bar_2  TimeSignature_4/4  NoteDensity_3
...
<eos>
```

Figure 3: Example description sequence, taken from von Rütte et al. (2022)

- Mean Pitch: This token contains a numerical value describing the mean of the pitch of all the notes played during the current bar. The mean value is quantized using pre-defined bins before being attached to the token.

- Mean Duration: Describes the mean note duration in the current bar using the same idea as the Mean Pitch token.

- Note Density: Describes the note density in the current bar using the same idea as the Mean Pitch token. The note density is the number of note onsets divided by the length of the current bar. Note onsets are time instants where at least one new note starts.

- Mean Velocity: Describes the mean note velocity in the current bar using the same idea as the Mean Pitch token. The velocity of a note is its loudness in the MIDI data, described on a scale from 0 to 127.

---

**Algorithm 1** ExpertDescription

---

**input** musical sequence $\mathbf{x}$
**output** description $\mathbf{d}$
  $\mathbf{d} \leftarrow ()$
  $b_1, \ldots, b_n \leftarrow \text{PARTITIONINTOBARS}(\mathbf{x})$
  **for** $b_i \in (b_1, \ldots, b_n)$ **do**
    $N \leftarrow \{n \mid n \text{ is a note with onset in } b_i\}$
    $I \leftarrow \{\texttt{inst} \mid \texttt{inst} \text{ is being played during } b_i\}$
    $C \leftarrow \{\texttt{chord} \mid \texttt{chord} \text{ is being played during } b_i\}$
    $q \leftarrow \text{duration of } b_i \text{ in quarter notes}$
    $\texttt{ts} \leftarrow \text{time signature at beginning of } b_i$
    $\texttt{nd} \leftarrow \frac{|N|}{q}$
    $\texttt{mp} \leftarrow \frac{1}{|N|} \sum_{n \in N} \text{PITCH}(n)$
    $\texttt{mv} \leftarrow \frac{1}{|N|} \sum_{n \in N} \text{VELOCITY}(n)$
    $\texttt{md} \leftarrow \frac{1}{|N|} \sum_{n \in N} \text{DURATION}(n)$
    Quantize $\texttt{nd}, \texttt{mp}, \texttt{mv}$ and $\texttt{md}$
    $d_i \leftarrow (i, \texttt{ts}, \texttt{nd}, \texttt{mp}, \texttt{mv}, \texttt{md}) \parallel \text{list}(I) \parallel \text{list}(C)$
    $\mathbf{d} \leftarrow \mathbf{d} \parallel d_i$
  **end for**
  **return** $\mathbf{d}$

---

Figure 4: Description calculation algorithm, taken from von Rütte et al. (2022)

## C   ALTERING THE DESCRIPTION

In our experiment, we alter the input description of FIGARO by changing one type of tokens at a time, over the whole sequence. For example, we calculate the description of an input MIDI file using the algorithm in Figure 4 and then pick a type of token to control, then apply the control procedure globally, over the whole sequence. Here are the different control procedures that we have tried:

- Shifts in Mean Pitch, Mean Velocity, or Mean Duration: We pick one of the three token types and shift its attached numerical value by a given value over the whole sequence. Shifts that result in out-of-bound values are clipped.

- Transposing the chords: Transposing chords is also a shift operation that shifts all the notes described by a chord by a given amount of half-steps. Half-steps are musical units for pitch.

- Removing an instrument: We pick an instrument randomly from the set of instruments used in the sequence, and then remove its corresponding token everywhere. Bars that do not involve the instrument remain untouched.

## D   EVALUATION METRICS

We evaluate the fidelity of the output to the attributes specified in the input description, and we also evaluate the overall musical quality of the outputs obtained from altered descriptions, compared to baseline outputs from unaltered descriptions.

### D.1   FIDELITY EVALUATION

We have three types of metrics:

- Macro Overlapping Area: This metric is used to evaluate the fidelity of pitch, velocity, and duration. The metric compares the similarity of two music sequence on the bar-level. It is calculated as follows:

  1. For each bar of each sequence, calculate the histogram of different values of the attribute of concern (Pitch, Velocity or Duration).
  2. Fit a Gaussian distribution to the histogram.

3. For each bar, calculate the overlapping area between the bar's Gaussian distribution and the Gaussian distribution of the corresponding bar from the other sequence.

4. Sum the overlapping areas over all bars

The metric is calculated between the output of FIGARO and the expected output. The expected output is the original MIDI data used for the input description, modified according to the control procedure (by shifting the velocity, duration, or pitch of all notes).

- Mean F1-score: Used for Instruments and Chords, as they can be considered as multi-label data. The comparison is also done using the expected output sequence.

- Normalized MSE: Used for Note Density. It is computed as the Mean Squared Error of the note density of each bar of the output against the specified density value in the input description, normalized by dividing by the note density of the input description.

## D.2   MUSIC QUALITY METRICS

These metrics measure the entropy of different aspects of a given piece of music (Wu & Yang, 2020).

- Pitch Histogram Entropy: Measures the entropy of the pitch of notes across predefined time blocks. We use 2 block sizes: 1 bar and 4 bars.

- Grooving Similarity: Measures the salience of rhythmic patterns.

- Chord Progression Irregularity: Measures the entropy of the used chords and the presence of patterns in the chord progression.

