# OpenReview forum: "Effects of Single-Attribute Control on the Music Generated by FIGARO"
_ICLR.cc/2023/TinyPapers — Submitted to Tiny Papers @ ICLR 2023_

### Official Review · Reviewer_ab4y · 2023-03-29

**Confidence:** 3

**Summary Of Contributions:**

This paper studies the relationship between individual musical attributes and the quality of FIGARO generated music. The results show that many musical attributes are correlated, and FIGARO maintains a consistent music quality when the individual attribute is controlled.

**Rating:**

Clear, Correct, and Reproducible (CCR): a submission which meets the reviewing criteria

**Strengths And Weaknesses:**

## Strengths and weaknesses

#### Clarity

1. Overall the paper is clear and easy to read.
2. How are the attributes represented in the experiment? For example, is the attribute "Chords" a number or a vector?

#### Correctness

1. The claims and conclusions are justified by the findings.

#### Reproducibility

1. The paper includes findings from an empirical experiment, but the code and data are not open-sourced.

#### Follows basic requirements

1. This paper follows the basic formatting requirements.

**Suggested Changes:**

## Suggested changes

1. What does "controlling the generated music" mean in Section 2 first paragraph?
2. It would be great if the paper could explain the columns of Table 1 in the main text.

---

> ### Author Response · Authors · 2023-05-30
> **Response to Reviewer ab4y**
>
> Dear Reviewer,
>
> Thank you for your feedback. We have taken all comments into consideration and uploaded a new paper revision.
>
> Regarding your comments:
> 1. _What does "controlling the generated music" mean in Section 2 first paragraph?_
> It means “conditionally generating music through user-defined constraints”. We have removed this part in the new paper revision.
>
> 2. _It would be great if the paper could explain the columns of Table 1 in the main text._
> We have added the full names of the columns of the table in the caption. Unfortunately we could not elaborate more on how each attribute is calculated in the main text, due to the 2-page limit of the Tiny Papers. Evaluation metrics are thoroughly explained in the appendix.
>
> We hope that we have answered your questions. Please let us know if you have any further concerns.
>
> Best regards

---

### Official Review · Reviewer_pz3g · 2023-03-30

**Confidence:** 3

**Summary Of Contributions:**

This paper attempts to control the musical properties of music generated by the FIGARO model, while evaluating the success of the control and the overall musical quality.

**Rating:**

Clear, Correct, and Reproducible (CCR): a submission which meets the reviewing criteria

**Strengths And Weaknesses:**

**Strengths**
- The idea of this paper employing musical properties to control music generation is interesting
- In the experiment, the description of the data processing is sufficient

**Weaknesses**
- Authors should explain how this algorithm differs from conditional generative models
- In the method, the author should introduce how the music attributes are embedded in the FIGARO model.
- In the experimental part, the author should introduce the detailed relationship between various indicators and music properties.


**Suggested Changes:**

1) Authors should explain how this algorithm differs from conditional generative models
2) In the method, the author should introduce how the music attributes are embedded in the FIGARO model.
3) In the experimental part, the author should introduce the detailed relationship between various indicators and music properties.

---

> ### Author Response · Authors · 2023-05-30
> **Response to Reviewer pz3g**
>
> Dear Reviewer,
>
> Thank you for your feedback. We have taken all comments into consideration and uploaded a new paper revision.
>
> Regarding your comments:
> 1. _Authors should explain how this algorithm differs from conditional generative models_
> In the new revision, we have made sure to extensively explain how FIGARO works, and to refer to other conditional music generation models in the related work section. In short, the main difference between our algorithm that uses FIGARO, and other conditional models, is that the model input is fully interpretable because it consists of domain-specific features, and it also allows fine-grained control over different parts of the sequence.
>
> 2. _In the method, the author should introduce how the music attributes are embedded in the FIGARO model._
> We have added a new appendix to specifically explain the architecture of FIGARO and how it embeds music attributes.
>
> 3. _In the experimental part, the author should introduce the detailed relationship between various indicators and music properties._
> We have added a new appendix to explain this.
>
>
> We hope that we have answered your questions. Please let us know if you have any further concerns.
>
> Best regards

---

### Official Review · Reviewer_t6rs · 2023-04-02

**Confidence:** 4

**Summary Of Contributions:**

The paper studies the effect of controlling the musical attributes of the music generated by the FIGARO and which music features contribute more to the music quality when controlled like instruments, chords , mean pitch etc

**Rating:**

High Potential (HP): a submission which meets the reviewing criteria and has potential to make an impact on the field

**Strengths And Weaknesses:**

Strengths
1. The research questions are clear and consistent with the experimental design and findings
2. The methodology and the controlled musical attributes are easy to describe and reproduce (did not see the code)
3. I like how figure 1 is explained clearly and is intuitive for a person who is not familiar to the music technical terms

Weaknesses
1. The correlations were not easily interpretable, would like to see some qualitative examples or descriptions of what they actually mean


**Suggested Changes:**

1. Examples of the descriptions used for control would be helpful for the reader to visualize the task
2. It will be helpful to introduce the relationship between the musical attributes and the music generated to better understand the correlations mentioned in the paper
3. I suggest the authors had some background literature on controlled music generation to better position the study in the field

---

> ### Author Response · Authors · 2023-05-30
> **Response to Reviewer t6rs**
>
> Dear Reviewer,
>
> Thank you for your feedback. We have taken all comments into consideration and uploaded a new paper revision.
>
> Regarding your comments:
>
> 1. _Examples of the descriptions used for control would be helpful for the reader to visualize the task_
> In the appendix of the new revision, we explain how descriptions work, how they are calculated, their vocabulary and the relationship of each token to the musical attributes. We also explain how each token is altered in order to control music attributes.
>
> 2. _It will be helpful to introduce the relationship between the musical attributes and the music generated to better understand the correlations mentioned in the paper._
> We have added this information in the appendix. We provide the pseudocode that calculates the music attributes from the original FIGARO paper, and we also explain the meaning of each attribute, which clarifies what changes are expected in the output sequence when altering a specific input attribute.
>
> 3. _I suggest the authors had some background literature on controlled music generation to better position the study in the field_
> We rewrote the first part of our literature review section to specifically focus on conditional music generation.
>
> We hope that we have answered your questions. Please let us know if you have any further concerns.
>
> Best regards

---

### Author Response · Authors · 2023-05-30
**New paper revision and archival**

Dear ICLR Program Chairs,

We have uploaded a new revision of our paper and have replied to all reviewers individually.

We would also like to state that we want to opt-in for the archival of our paper.

Best regards

---

### Comment · Area_Chair_1r9v · 2023-06-06
**Archival Criterion Check**

This work meets the threshold for archival, contents the URM statement and is deanonymized.

---

### Meta-Review · Area_Chair_1r9v · 2023-04-08

**Recommendation:** Invite to present
**Confidence:** 4

**Metareview:**

The pros and cons summarized from the reviewers are following:

Pros:

1. Clear research questions and alignment with experimental design and findings.
2. Methodology and controlled musical attributes are easily understandable and reproducible.
3. Figure 1 is well-explained and intuitive for readers not familiar with music technical terms.
4. The paper is clear and easy to read.
5. Claims and conclusions are justified by the findings.
6. The paper follows the basic formatting requirements.

Cons:

1. The correlations were not easily interpretable, and there is a need for more qualitative examples or descriptions.
2. The paper lacks details on how the algorithm differs from conditional generative models.
3. Insufficient information on how the music attributes are embedded in the FIGARO model.
4. The experimental part needs more details on the relationship between various indicators and music properties.
5. Code and data are not open-sourced, affecting reproducibility.

**Summary:**

The paper investigates the impact of controlling musical attributes on the quality of music generated by the FIGARO model. It explores the relationship between individual attributes, such as instruments, chords, and mean pitch, and the overall musical quality while maintaining consistency in the generated music.

**Comments And Feedback To The Authors:**

As summarized from the reviewers' comments, the main areas of concern include:

1. Lack of easily interpretable correlations: The paper needs to provide more qualitative examples or descriptions to make the correlations more understandable for the readers.
2. Insufficient details on algorithm differentiation: The authors should clarify how their algorithm differs from existing conditional generative models to emphasize the originality of their approach.
3. Incomplete information on embedding music attributes: The paper should provide more information on how music attributes are embedded in the FIGARO model to improve the clarity of the methodology.
4. Limited information in the experimental part: The paper should include more details on the relationship between various indicators and music properties to strengthen the experimental results.

By addressing these issues, the paper would have a stronger foundation and could potentially receive a higher recommendation from the reviewers.

**Reason For Not Giving A Higher Recommendation:**

The reviewers pointed out that the correlation between attributes and quality is not easily interpretable, necessitating further analysis. Moreover, the limited information on how the algorithm differs from other conditional generative models also constrains the reproducibility of the work. Addressing these issues would strengthen the paper's overall quality and impact.

**Reason For Not Giving A Lower Recommendation:**

N/A

---

> ### Author Response · Authors · 2023-05-30
> **Response to Area Chair 1r9v**
>
> Dear ICLR Area Chair,
>
> Thank you for your feedback and meta-review. We have taken all comments into consideration and uploaded a new paper revision.
>
> Regarding your comments:
> 1. _The correlations were not easily interpretable, and there is a need for more qualitative examples or descriptions._
> We have added a thorough description of all musical attributes and how we control them. We have also linked a Github repository that contains some examples of generated music.
>
> 2. _The paper lacks details on how the algorithm differs from conditional generative models._
> In the new revision, we have made sure to extensively explain how FIGARO works, and to refer to other conditional music generation models in the related work section. In short, the main difference between our algorithm that uses FIGARO, and other conditional models, is that the model input is fully interpretable because it consists of domain-specific features, and it also allows fine-grained control over different parts of the sequence.
>
> 3. _Insufficient information on how the music attributes are embedded in the FIGARO model._
> We have added a new appendix to specifically explain the architecture of FIGARO and how it embeds musical attributes to generate music.
>
> 4. _The experimental part needs more details on the relationship between various indicators and music properties._
> We added a new appendix to go through each evaluation metric and its relationship with the musical attributes.
>
> 5. _Code and data are not open-sourced, affecting reproducibility._
> We have linked to a Github repository with all the code and links to the dataset.
>
>
> We hope that we have answered your questions. Please let us know if you have any further concerns.
>
> We would also like to explicit state that we want to opt-in for the archival of our paper.
>
> Best regards

---

### Decision · Program_Chairs · 2023-04-08

Invite to present